# Epigenetic Regulation of Plant Gametophyte Development

**DOI:** 10.3390/ijms20123051

**Published:** 2019-06-22

**Authors:** Vasily V. Ashapkin, Lyudmila I. Kutueva, Nadezhda I. Aleksandrushkina, Boris F. Vanyushin

**Affiliations:** Belozersky Institute of Physico-Chemical Biology, Lomonosov Moscow State University, Moscow 119234, Russia; kutueva@genebee.msu.ru (L.I.K.); aleks@genebee.msu.ru (N.I.A.); vanyush@belozersky.msu.ru (B.F.V.)

**Keywords:** gametophyte, plant development, epigenetics, gene expression, DNA methylation, chromatin, siRNA

## Abstract

Unlike in animals, the reproductive lineage cells in plants differentiate from within somatic tissues late in development to produce a specific haploid generation of the life cycle—male and female gametophytes. In flowering plants, the male gametophyte develops within the anthers and the female gametophyte—within the ovule. Both gametophytes consist of only a few cells. There are two major stages of gametophyte development—meiotic and post-meiotic. In the first stage, sporocyte mother cells differentiate within the anther (pollen mother cell) and the ovule (megaspore mother cell). These sporocyte mother cells undergo two meiotic divisions to produce four haploid daughter cells—male spores (microspores) and female spores (megaspores). In the second stage, the haploid spore cells undergo few asymmetric haploid mitotic divisions to produce the 3-cell male or 7-cell female gametophyte. Both stages of gametophyte development involve extensive epigenetic reprogramming, including siRNA dependent changes in DNA methylation and chromatin restructuring. This intricate mosaic of epigenetic changes determines, to a great extent, embryo and endosperm development in the future sporophyte generation.

## 1. Introduction

Plants have a complex life cycle with alternating diploid (sporophyte) and haploid (gametophyte) generations (Figure 1) [1]. Unlike animals, plants do not have a specialized germline cell lineage in the developing embryo. Instead, the reproductive lineage cells differentiate from somatic tissue of adult plants. In flowering plants, the male gametophyte (pollen) develops within the anther and the female gametophyte (embryo sac)—within the ovule. During flower development, cells in the anther primordium divide and differentiate to form several cell types, including pollen mother cells (PMCs). At the early stages of pollen development, PMCs undergo meiotic divisions to form haploid microspore cells. Unlike in animals, gamete formation in plants involves meiotic and subsequent haploid mitotic divisions. In each microspore, the nucleus first migrates to one side of the cell. An asymmetric mitotic division produces a large vegetative cell (VC) and a small generative cell (GC). The VC completely encloses the GC, which undergoes a second mitotic division to produce two sperm cells (SCs). This second mitotic division can occur either in the anther or during pollen tube growth. The SCs are connected with the VC nucleus by the generative cell plasma membrane, forming the male germ unit. This structure ensures the simultaneous delivery of both SCs to the ovule. Upon pollination, the VC forms a pollen tube, which grows towards female tissues to deliver the SCs into the female gametophyte.

The female gametophyte, also known as embryo sac, develops within the ovule [1,2]. A region called the nucellus gives rise to a single pre-meiotic cell—the megaspore mother cell (MMC). MMC undergoes two meiotic divisions resulting in four haploid megaspores. In most plants, only one megaspore survives, while the remaining three undergo programmed cell death. There are many variations in cell numbers and organization of the embryo sac in angiosperms, but the vast majority of flowering plants have the monosporic 8-nucleate Polygonum-type embryo sacs, containing seven cells of four distinct cell types [3]. In these plants, the megaspore undergoes three sequential haploid mitotic nuclear divisions resulting in the generation of eight nuclei that migrate and contribute to the mature embryo sac. Then cellularization occurs, resulting in the formation of seven cells since two nuclei become the homo-diploid central cell nucleus. The central cell (CC) will give rise to the endosperm. The egg cell (EC) will give rise to the embryo, two synergid cells (SyCs) serve for pollen tube attraction and entry, and three antipodal cells (ACs) have unknown function. The embryo sac is enclosed within diploid sporophytic integument tissues, which will become the seed coat. At fertilization, the pollen tube enters into the embryo sac through the opening in the integuments at the distal (micropylar) end of the ovule, adjacent to the EC and two SyCs. The proximal (chalazal) end of the ovule is adjacent to the three ACs. The homo-diploid CC is the largest cell of the embryo sac.

The mitotic divisions of megaspore leading to the formation of embryo sac are strictly position-dependent and ordered in time. The first division produces two nuclei that migrate to opposite poles of the embryo sac. The second and third divisions give rise to four nuclei at each pole. Their positioning correlates with future cell specification. At the micropylar pole, the two most distal nuclei form the SyCs; of the two remaining nuclei, a more distal one forms the EC, while the more central one becomes one of the two nuclei of the CC. Of the four nuclei at the chalazal pole, the most centrally located one becomes the other nucleus of the CC. It migrates towards the micropylar pole and stays close to the first nucleus of the future CC. In *Arabidopsis*, two nuclei of the CC fuse just before the fertilization to form the homo-diploid CC nucleus, while the remaining three nuclei at the chalazal pole form the ACs that will disappear at the time of fertilization. The overall development of the female gametophyte is essentially the same in cereals, except for a few notable differences. The first is that two polar nuclei remain unfused until fertilization when they fuse with the sperm nucleus to form the triploid endosperm. The second is that the three ACs do not degenerate, but instead begin to proliferate, resulting in up to 40 ACs in maize. In cereals, ACs resemble cells involved in nutrient transfer; it was suggested that these cells might play an important role in the transfer of nutrients from maternal tissues to the cereal endosperm [4]. However, in the majority of flowering plants, the ACs degenerate before fertilization, and their function is still unknown.

The *eostre* mutation in *Arabidopsis* leads to the switch of a SyC into an extra EC [5]. In maize, mutation of *indeterminate gametophyte1* results in the supernumerary nuclei in the embryo sac that assume specific cell fates correlated with their position [6]. Positional information within the embryo sac may be determined by the gradient distribution of some morphogenetic factor(s), similar to those in animal embryos. It has been shown that auxin distribution could serve as such morphogenetic factor [7]. At the earlier stages of female gametophyte development, the auxin signal is strong in the nucellus. At later stages, a high auxin concentration is detectable at the micropylar pole of the embryo sac up to the third mitotic division and eight nuclei positioning before cellularization. Thus, the specification of SyCs occurs in nuclei exposed to the highest levels of auxin, while the specification of ACs occurs in nuclei at the lowest auxin concentrations; intermediate concentrations lead to the EC and CC formation.

Auxin appears to be the primary determinant of cell specification in developing female gametophyte, while the maintenance of cell fates has been found to depend on inter-cellular interactions between the EC and CC on the one side, and the SyCs and ACs, respectively, on the other side. Mutants of some genes (*LIS*, *GFA1/CLO*, and *ATO*) have been found to change the cell identities inside the embryo sac after their normal specification [8,9]. A mechanism of lateral inhibition by the EC and CC has been suggested to prevent the excessive gamete cell formation and maintain the identity of accessory cells [8].

In flowering plants, both SCs in the pollen are used to fertilize cells of the female gametophyte—a unique process of double fertilization. When pollen lands on the stigma, the pollen tube penetrates the embryo sac through the micropyle towards the SyCs that secrete the attracting signal, and bursts releasing the sperm cells. The programmed death of the receptive SyCs follows while the SCs fertilize the EC and CC. The fertilization of the EC cell results in the diploid zygote, which will initiate the embryo of the next sporophyte generation. The fertilization of the CC results in the triploid endosperm that functions as a source of nutrients for the embryo development and the seedling germination. From the moment of pollen germination until sperm discharge into the embryo sac, the pollen tube communicates with at least five different sporophytic and three different gametophytic cell types [10]. Its growth is regulated by much different guidance, attraction, and support mechanisms.

## 2. Basics of Epigenetic Regulation in Plants

### 2.1. DNA Methylation

The most thoroughly studied mechanism of epigenetic regulation in plants is DNA methylation [11,12]. In the *Arabidopsis* genome, 24% of the symmetric CG sites, 6.7% of the symmetric CHG sites, and 1.7% of the asymmetric CHH sites (where H is any nucleotide except G) were shown to be methylated [13]. Considering the genome frequency of these target sites, about 55% of all methylated cytosine residues are in CG, about 23% are in CHG, and about 22% are in CHH sites. All three types of the methylated sites are present in repeat and transposable elements (TEs) sequences in pericentromeric heterochromatin regions and small heterochromatic patches interspersed in euchromatin regions, whereas mostly CG sites are methylated in protein-coding gene sequences [13,14]. DNA methylation in plants is carried out by a large set of specific DNA methyltransferases, some of which have no analogs in animals [11,12]. These enzymes are classified into de novo (those that methylate previously unmethylated target sites) and maintenance (those that maintain methylation of previously methylated target sites) DNA methyltransferases. Maintenance methylation of CG sites in all genome compartments is carried out by DNA METHYLTRANSFERASE 1 (MET1) in cooperation with three members of the VARIANT IN METHYLATION protein family, VIM1–VIM3, that recognize hemimethylated CG sites in the newly replicated DNA via a special SET and RING associated (SRA) domain [15]. The plant-specific CHROMOMETHYLASE 3 (CMT3) maintains the methylation of symmetric CHG sites and probably contributes to the methylation of asymmetric CHH sites. Unlike MET1, CMT3 has no internal preference for the hemimethylated sites. CMT3 recognizes, via its chromodomain, nucleosomes containing the histone H3 molecules methylated at the ninth lysine residue (H3K9me1, H3K9me2, and H3K9me3) [16], whereas the H3K9-specific methyltransferases KRYPTONITE (KYP), also known as SU(VAR)3-9 HOMOLOG 4 (SUVH4), SUVH5, and SUVH6 recognize via their SRA domains the DNA loci methylated at sites of any type with a clear-cut preference for CHG and CHH [17]. Thus, the CMT3 maintenance activity *in vivo* is a consequence not of its substrate preferences, but rather of the self-reinforcing loop between CMT3 and H3K9-specific methyltransferases. De novo DNA methylation at all sequence contexts is carried out by DOMAINS REARRANGED METHYLTRANSFERASE 2 (DRM2) [18], with a modest contribution of DRM1 in the egg cell of the female gametophyte and at earlier stages of embryo development [19]. Since only one strand of DNA contains a C residue at CHH sites, their methylation could be maintained by repeated de novo methylation only. For a long time, the DRM2 methyltransferase has been regarded to be responsible for all CHH methylation [11,12]. Recently an alternative CMT2-dependent pathway was shown to be responsible for a significant part of CHH methylation in the *Arabidopsis* genome [20]. Moreover, DRM2 and CMT2 have distinct target preferences: CMT2 mostly methylates long TEs in the heterochromatic pericentromeric regions, whereas DRM2—short TEs in euchromatic chromosome arms. Similar to CMT3, CMT2 depends on H3K9 methylation, but unlike CMT3, which preferentially methylates symmetric CHG sites, CMT2 has a clear-cut preference for asymmetric CHH sites [21]. Besides, CMT2 preferentially binds to H3K9me2 and H3K9me3 and much weaker to H3K9me1, whereas CMT3 binds all three H3K9me forms equally well. Thus, the distribution of these histone marks significantly affects CMT2 and CMT3 targeting to different genome loci.

Target sites to be de novo methylated are recognized with the aid of 24 nucleotide small interfering RNAs (24 nt siRNAs), complementary to parts of the respective loci, in a complex multi-stage pathway RNA-directed DNA methylation (RdDM) [22,23]. At the beginning of the “classic” RdDM pathway, a plant-specific RNA polymerase IV (Pol IV) is recruited, by SHH1 protein that recognizes unmethylated H3K4 and methylated H3K9 through its Tudor domain, to a subset of its possible target loci to generate short (26–45 nt) single-stranded RNA (ssRNA) transcripts. These RNA transcripts are subsequently used as templates by Pol IV-associated RNA-DEPENDENT RNA POLYMERASE 2 (RDR2) to generate double-stranded RNAs (dsRNAs), which are then cut into double-stranded 24 nt siRNAs by the endoribonuclease DICER-LIKE 3 (DCL3). These siRNAs are methylated at their 3′ ends by the methylase HUA ENHANCER 1 (HEN1), and one 24-nt strand is loaded into ARGONAUTE 4 (AGO4) or AGO6. Meanwhile, another plant-specific RNA polymerase V (Pol V) transcribes the target loci to be DNA methylated resulting in the production of non-coding transcripts that remain attached to their loci of origin and serve as scaffolds for AGO4/6-siRNA binding via sequence complementarity to 24 nt siRNAs. The AGO4/6-siRNA complex with Pol V transcript subsequently recruits the DRM2 methyltransferase, resulting in de novo cytosine methylation of the adjacent DNA. Following RdDM, heterochromatin is formed through the recruitment of histone modifying enzymes, such as histone deacetylase 6, H3K9-specific methyltransferases KYP (SUVH4), SUVH5, and SUVH6, and H3K4-specific demethylase JUMONJI 14 (JMJ14). Several thousand sites of Pol IV and Pol V occupancy have been identified, but consensus promoter motifs were not found.

Interestingly, the final stages of RdDM are dependent on the SRA domain-containing proteins SUVH2 and SUVH9 that recognize methylated DNA sites [24]. Paradoxically, the classic RdDM pathway of de novo DNA methylation appears to be essentially maintenance one, since previously methylated DNA sequences are preferentially targeted to be further methylated. The classic RdDM pathway mainly acts as a positive backward loop reinforcing methylation states of silenced TEs in heterochromatin regions. An interesting question is how originally unmethylated DNA loci could be methylated. Until fairly recently it was generally believed that two major classes of siRNA participate in different modes of gene silencing: 24 nt siRNAs in transcriptional gene silencing (TGS) via RdDM, whereas 21−22 nt siRNAs in post-transcriptional gene silencing (PTGS) via targeting complementary mRNAs and inducing their cleavage and/or inhibiting translation [25]. An alternative RdDM pathway has been uncovered that may fulfill the de novo function, as it starts with transcripts produced by RNA polymerase II (Pol II) [26]. Since this alternative pathway involves RDR6 instead of RDR2, it is usually called RDR6-RdDM. It also involves DCL2 or DCL4 known to produce 21–22 nt siRNAs in the PTGS pathway. The Pol II produced transcripts of active TEs are converted by RDR6 to dsRNA molecules that are then cleaved into 21–22 nt siRNAs by DCL2 or DCL4. The 21–22 nt siRNAs are loaded into PTGS complexes with AGO1 or could be rooted via AGO6 or AGO2 complexes to Pol V-DRM2 dependent RdDM. The RDR6-RdDM pathway regulates only a few individual TEs in the TE-silenced epigenome of wild-type Col plants, whereas the number of such TEs is much larger when TEs are transcriptionally active in the *ddm1* mutant plants. Only 15 of 318 TE subfamilies were found to be the source of all the RDR6-dependent TE siRNAs in the *ddm1* mutant epigenome, most of them from Athila family retrotransposons. However, these 15 TE subfamilies account for about a quarter of the total TEs length and nearly 5% of the entire *Arabidopsis* genome. The RDR6-RdDM pathway could act upstream of canonical RdDM as an initial step of TEs methylation and silencing [26,27]. However, since RdR6-RdDM requires Pol V, it should be suggested that Pol V can be recruited to unmethylated DNA loci.

In plants, methylated cytosine (5mC) residues could be actively removed from DNA by dedicated 5mC-specific DNA glycosylases REPRESSOR OF SILENCING 1 (ROS1), DEMETER (DME), DEMETER-LIKE 2 (DML2), and DEMETER-LIKE 3 (DML3) via a base excision repair (BER) pathway [28]. The gene encoding a DNA demethylase DME was discovered as normally expressed in the CC of the female gametophyte and needed for activation of the maternal allele of an imprinted gene *MEDEA* (*MEA*) [29]. It also participates in maternal imprinting of other genes, such as *FLOWERING WAGENINGEN* (*FWA*), *FERTILIZATION INDEPENDENT SEED 2* (*FIS2*) and *MATERNALLY EXPRESSED PAB C-TERMINAL* (*MPC*) [28]. Patterns of DNA methylation are maintained by the equilibrium between DNA methylation and demethylation. It has been shown that multiple genomic loci targeted by RdDM are also targets of active demethylation by ROS1 [30]. Relative to the general TE population, ROS1 targets TEs that are near protein-coding genes. It was suggested that ROS1 prevents the spreading of DNA methylation from these TEs, known to be the main targets of RdDM, to nearby genes. Importantly, thousands of 24 nt siRNA target loci gain methylation in *ros1* mutant plants, indicating that ROS1 actively prevents their methylation in wild-type plants. Also, ROS1 antagonizes RdDM-independent DNA methylation at more than a thousand loci. ROS1 is recruited to its target loci by a multiprotein complex containing methyl-CpG-binding domain (MBD) protein 7 (MBD7) and increased DNA methylation 1 (IDM1), IDM2, and IDM3 [31]. It was shown that a helitron family TE proximal to the *ROS1* gene is also targeted to both RdDM and DNA demethylation [32,33]. Unexpectedly, the expression of *ROS1* appeared to be stimulated by methylation and inhibited by demethylation of a sequence between TE and 5′ untranslated region (UTR) of the *ROS1* gene. It was hypothesized that this sequence (methylation monitoring sequence—MEMS) serves as an epigenetic rheostat (methylstat) that fine-tunes the DNA demethylase activity to an optimal level for ensuring both robust TEs silencing and transcriptional activity of nearby genes. There are two additional members of the *DME* gene family in *Arabidopsis* genome, namely *DML2* and *DML3*. The protein encoded by these genes also appeared to be 5mC-specific DNA glycosylases involved in the maintenance of correct methylation of multiple genes [34,35].

### 2.2. Chromatin Modification

Epigenetic marks that are highly conserved in many eukaryotic organisms including plants consist of modifications on histone tails. Different histones are covalently modified at different lysine and arginine residues by methylation, acetylation, ubiquitination, phosphorylation, and other chemical marks [11,36]. Three types of histone methylation, histone H3K4 mono/di/tri-methylation (H3K4me1, H3K4me2, and H3K4me3), histone H3K27 tri-methylation (H3K27me3) and histone H3K9 di-methylation (H3K9me2), are the most well-studied in plants. In *Arabidopsis*, all three types of H3K4me marks were found to be located almost exclusively at genes and promoters, two-thirds of genes containing at least one type of such marks [37]. H3K4me2 and H3K4me3 show predominant promoter localization, while H3K4me1 is distributed across gene bodies. H3K4me3-containing genes are highly expressed in various tissues. H3K4me2/3 and DNA methylation are mutually exclusive in the same gene loci, whereas H3K4me1 is highly correlated with CG DNA methylation in the gene bodies.

H3K27me3 is widely known as a major repressive mark for gene expression. In *Arabidopsis*, it regulates a large number (about 4400) of genes, and its maintenance appears to be mostly independent of other epigenetic pathways [38]. Different from animals, in plants H3K27m3 is mostly restricted to the transcribed regions of genes, including multiple genes coding for transcription factors. H3K27me3-regulated genes are usually expressed at low levels and with a high degree of tissue specificity. These features indicate an important role of H3K27me3 in epigenetic regulation of plant development. *Arabidopsis* has three homologs of Enhancer of zeste (E(z))—a SET (Su(var)3-9, Enhancer of zeste, Trithorax) domain histone methyltransferase specific for H3K27 tri-methylation that is a key subunit of polycomb repressive complex 2 (PRC2). The plant chromodomain protein LHP1 (LIKE HETEROCHROMATIN PROTEIN 1) has been found to colocalize with H3K27me3 genome-wide and to bind H3K27me3 in vitro [39]. LHP1 and some other proteins probably function as H3K27me3 readers in plants to translate this mark into various functional outcomes.

H3K9me2 is the pre-dominant mark of heterochromatin in plants. As was already noted above, H3K9me2 plays a crucial role in non-CG DNA methylation by chromomethylases CMT3 and CMT2. Not unexpectedly, a very high coincidence between H3K9m2 and CHG methylation was observed throughout the *Arabidopsis* genome, whereas CG-methylated genes did not contain H3K9m2 [40]. Two major patterns of H3K9m2 distribution were detected. TEs and other repeat elements in the euchromatic arms contain small patches of H3K9m2 at relatively low levels, whereas pericentromeric and centromeric regions contained long blocks of H3K9m2 at much higher levels. Thus, a complex cross-talk between H3K9m2 and non-CG DNA methylation exists in different genome compartments.

### 2.3. Small Non-Coding RNAs

Plants have three groups of small RNAs that differ in their biogenesis and function: 20–22 nt microRNAs (miRNAs) produced by DCL1, 21–22 nt siRNAs produced by DCL4 and DCL2, and 24 nt siRNAs produced by DCL3 [25]. All these small RNAs are produced via cleavage of dsRNAs by DCL endoribonucleases and exert their respective functions via recognition of complementary RNAs as part of RNA-induced silencing complexes (RISCs) with AGO family proteins, resulting in AGO-mediated degradation or translational inhibition of RNA (PTGS), or transcriptional silencing of associated homologous DNA (TGS). All plant small RNAs are stabilized and protected from degradation through 2′-*O*-methylation by HEN1.

The precursors of miRNAs are single-stranded RNA transcripts with imperfect hairpin structures of non-protein coding microRNA genes (*miR*s). The precursors of siRNAs are dsRNAs, which could be produced by transcription of inverted repeats, annealing of complementary sense- and antisense transcripts, or RDR dependent copying of single-stranded RNA templates. miRNAs play a crucial role in post-transcriptional regulation of gene expression in plants development. Mature miRNA associates with AGO1 in the nucleus and the AGO1-miRNA complex moves to the cytoplasm, where it exerts its PTGS action via mRNA cleavage and/or translational repression. Most miRNAs are 21 nt in length and associated with AGO1. The mRNA cleavage by 21 nt miRNA-AGO1 complexes results in fragments that are rapidly degraded. Transcripts that are cleaved by 22 nt miRNA-AGO1 complexes or recognized at two sites by 21 nt miRNA-AGO7 complexes could be not degraded, and become substrates for RDR6. Their copying by RDR6 and subsequent processing by DCL2 or DCL4 results in the production of secondary siRNAs that are regularly “phased” relative to the miRNA cleavage site in the target RNA. Therefore, these secondary siRNAs are called phasiRNAs. Trans-acting siRNAs (tasiRNAs) are 21 nt phasiRNAs produced via DCL4 cleavage of non-coding tasiRNA-generating (TAS) transcripts. Most tasiRNAs are incorporated into AGO1 to silence endogenous mRNAs during plant development. Besides TAS loci, there are numerous phasiRNA-producing loci in plant genomes, including those in protein coding sequences.

Many 21 and 22 nt siRNAs are involved in PTGS by targeting and degrading transcripts of virus genomes, TEs or transgenes. These transcripts do not possess a 5′-end cap or 3′-end poly(A) tail, rendering them to become templates for RDR6 to produce dsRNAs. These dsRNAs are cleaved by DCL2/4 into 21–22 nt siRNAs that are loaded into AGO1-containing PTGS complexes.

*Arabidopsis* has 10 *AGO* family genes that form three phylogenetic clades: *AGO1/5/10*, *AGO2/3/7*, and *AGO4/6/8/9* [41]. *AGO8* is widely considered to be a pseudogene since it contains a splicing-inducing frameshift, probably leading to the formation of a nonfunctional protein.

As was noted above, AGO1 is the effector protein for miRNAs and tasiRNAs and also for 21–22 nt siRNAs involved in PTGS. As an effector of miRNAs, AGO1 participates in most developmental pathways in plants. Of all *Arabidopsis* AGO proteins, AGO10 has the highest sequence similarity to AGO1. It is known to participate in stem cell maintenance in shoot apical meristem and leaf development [42]. Another member of the AGO1 clade, AGO5, is expressed in and around the developing megaspores during ovule development [43]. Unlike other members of the clade, AGO5 preferentially binds 24 nt siRNAs [42,43].

All members of AGO2 clade possess slicer activity [42]. As was already noted, AGO7 is loaded with 21 nt miRNAs for dual site targeting of non-coding transcripts in phasiRNA biogenesis. Besides its participation in RDR6-RdDM pathway noted above, AGO2 is considered to play a role in 21 nt siRNA-mediated repair of double-stranded DNA breaks (DSBs) [42]. AGO3 has the maximal sequence similarity to AGO2. Interestingly, unlike AGO2, it binds 24 nt siRNAs and may be functionally redundant with AGO4 in mediating the TGS at some loci [44].

The members of AGO4 clade, AGO4, AGO6, and AGO9 predominantly bind DCL3-dependent 24 nt siRNAs mediating the “classic” RdDM [42]. Their functional divergence is determined by differences in their expression patterns in plant tissues and interaction with target loci [45]. AGO4 is widely expressed throughout the embryo and mature tissues, while AGO6 and AGO9 are expressed at lower levels in the embryo and meristematic regions. Thus, AGO4 is the main AGO-effector in “classic” RdDM pathway. AGO4 and AGO6 were found to have different co-localization with RNA polymerases [46]. AGO4 co-localizes with Pol II in the nucleoplasm and with Pol V in perinucleolar foci, while AGO6 co-localizes with Pol V in the nucleoplasm and appears to be more important in accumulation of Pol V-dependent transcripts. Thus, AGO4 and AGO6 are functionally divergent in the “classic” 24 nt siRNA-dependent RdDM pathway. Furthermore, as was already noted, AGO6 participates in 21–22 nt siRNA-dependent RDR6-RdDM pathway.

Thus, small RNA-dependent silencing pathways in flowering plants are broadly diversified systems that have been evolved based on gene duplication and divergence to fulfill variable functions of genome protection and developmental regulation.

This section is not intended to be a comprehensive review of epigenetic regulation in plants. An interested reader may find the intriguing details of plant epigenetics in available reviews [11,12,22,23,25,28,36,41,42].

## 3. Epigenetic Regulation at the Meiotic Stage of Gametophyte Development

PMCs constitute a small share (about 1%) of anther tissue in *Arabidopsis*, making their isolation in a pure form very difficult. By using original manual methods of PMCs isolation, two groups succeeded in collecting enough pure PMCs for RNA-sequencing analyses of transcriptomes and found more than 20,000 genes to be expressed in PMCs [47,48]. Less genes functionally connected with DNA and RNA metabolism and more genes of signal transduction pathways are expressed in PMCs compared with total anthers and seedlings. About 700 genes showed at least two-fold difference in expression levels between pure PMCs and total anthers, most of them (>600) being preferentially expressed in PMCs [47]. A large genomic block on chromosome II pericentromeric region known to be of mitochondrial genome origin contained 152 genes, of which 100 genes were preferentially (55 genes) or specifically (45 genes) expressed in PMCs compared with anthers. More than 1000 putative transcription factor (TF) genes were expressed in PMCs, indicating a diverse transcriptional regulation landscape of meiotic cells [48]. A great number of transposable elements (TEs) were specifically expressed in PMCs compared with anthers and seedlings. Of 1117 TEs that showed at least a two-fold difference in expression levels between PMCs and anthers, 871 TEs were expressed in PMCs only and 18 TEs—in anthers only [47]. Of 228 TEs that were expressed in both PMCs and anthers, 165 showed higher expression in PMCs and 63—in anthers. Among the 1223 TEs that were differentially expressed between PMCs and seedlings, 1148 TEs were expressed in PMCs only and 17 TEs—in seedlings only. Thus, a total of 1271 TEs are expressed in PMCs, which is more than 30% of all TEs known in *Arabidopsis*. Since the numbers of TEs expressed in anthers and seedlings are much smaller, TEs could play some unique role in meiosis. The *AGO9* gene is preferentially expressed in anthers compared with PMCs in accordance with its postmeiotic function [47]. In contrast, *AGO3* and *AGO8* are preferentially or specifically expressed in PMCs. Hence, gene silencing and DNA methylation genes are differently regulated at meiotic and postmeiotic stages of male gametophyte development.

A laser capture microdissection analysis showed that *Arabidopsis* MMC transcriptome is enriched for transcriptional regulators and RNA helicases compared with surrounding nucellus cells. Mutations of the RNA-helicase gene *MNEME* (*mem*) lead to the production of multiple MMC-like cells and epigenetic perturbations in gametophyte nuclei [49]. Similarly, mutations in the *AGO9* gene lead to the formation in the ovule of ectopic MMCs that differentiate without undergoing meiosis into diploid megaspores and arrest at the one-nuclear stage [50]. Interestingly, the *AGO9* gene is expressed not in the MMC itself but the surrounding cells of the ovule. Thus, it appears to act non-cell-autonomously to limit excessive MMC formation. Mutants in other genes of the 24 nt siRNA silencing pathway, such as *RDR2*, *DCL3*, *NRPD1*, and *NRPE1*, have the same effect, indicating the MMC fate restriction to be mediated by RNA-directed silencing [14]. The primary function of AGO9 seems to be the siRNA-mediated silencing of transposons since gametes in *ago9* mutant plants exhibit active transcription of normally silent transposons. In wild-type Col and Ler ovules about 9.8% and 17.2%, respectively, contain two MMCs [51]. Thus, some ectopic gametophytic precursors are produced in *Arabidopsis* ecotypes. In heterozygous *ago9* mutants, about 21% ovules contain two or more MMCs, whereas, in homozygous *ago9* mutants, the number of such ovules reaches about 29%. Similar increases in ovules with excessive MMCs were observed in heterozygous and homozygous *ago4*, *ago6*, and, unexpectedly, in *ago8* mutants. Thus, all the clade *AGO4* genes are involved in the restriction of MMC specification, despite *AGO8* being considered a pseudogene.

In contrast, the frequency of ovules with excessive MMCs in homozygous *ago1* and *ago5* mutants was not increased compared with wild type. Interestingly, the analysis of *ago4 ago9* double mutants showed that the simultaneous absence of AGO4 and AGO9 tends to repress the production of excessive MMCs. Hence *AGO9* and *AGO4* genetically interact to restrict the production of excessive MMCs. In the *ago4* mutant, the expression of *AGO6* was equivalent to wild type, whereas, in the *ago9* mutant, the expression of *AGO6* was significantly increased. Despite being overexpressed in *ago9*, in the double mutant *ago4 ago9, AGO6* showed normal expression, while *AGO8* was not affected in *ago4* and *ago9* single mutants, but was significantly increased in the *ago4 ago9* double mutant. Thus, interactions between *AGO4*-clade members at the level of transcriptional activity appear to present a robust, partially redundant mechanism to restrict MMC production to one cell per ovule.

MMC undergoes significant chromatin changes, including a global reduction in heterochromatin, loss of the linker histone H1, and changes in histone modifications [52]. The histone H1 depletion is detectable at the early stages of MMC differentiation, when morphological differentiation is practically absent. Apparently, changes in chromatin structure determine the transition from somatic to reproductive phenotype in this cell. Later, further chromatin remodeling and biphasic changes in histone modifications occur in the MMC. Notably, despite very low heterochromatin content in MMC, the remaining heterochromatin is highly enriched in H3K9me2. One aspect of nucleosome remodeling is the dynamic turnover of the centromere-specific histone H3 variant (CENH3) and the incorporation of a specific histone H3.3 variant—HTR8.

Further chromatin changes involve an increase in the permissive histone mark H3K4me3 and the reduction of repressive mark H3K27me3. These changes are consistent with the establishment of a permissive chromatin environment. However, the reduced levels of permissive histone marks H3K4me2 and H4K16ac and of active (Ser2-phosphorylated) RNA polymerase II indicated a limited transcriptional competence of the MMC. The rebuilding of heterochromatin occurs at meiosis via gain of H1 and H3K9me2 methylation. After meiosis, the megaspore undergoes a second wave of heterochromatin reduction and loss of H1 and H3K9me2 [52], while the production of gametophyte syncytium nuclei is accompanied by H3K9me2 gain [53].

The wide expression of TEs in PMCs noted above [47,48] may suggest that global decondensation of heterochromatin loci occurs in these cells, similar to MMCs. A dynamic turnover of CENH3 was detected in *Arabidopsis* PMCs [54], further supporting this view. Indeed, changes in nuclear morphology of differentiating PMC, marked by enlargement of nuclei and nucleoli, were observed in *Arabidopsis* anthers [55]. Quantitative analyses revealed a 5-fold increase in nuclear volume, a 47.9% decrease in heterochromatin content and reduction in the mean number of distinct chromocenters from 7 to 3 in average in PMC, compared with epidermal cells in the anther wall. Thus, chromatin decondensation occurs in PMC similar to that in MMC. Also, eviction of the linker histone H1 and a reduction of H2A.Z (a histone variant antagonizing DNA methylation and chromatin compaction) were observed, similar to the situation in MMC. However, a large quantity of H2A.Z was detected in the microspores, indicating that H2A.Z is reloaded at or shortly after meiotic divisions. In accordance with the permissive chromatin configuration established in PMC, decreased repressive histone marks H3K27me1 and H3K27me3 and increased permissive histone mark H3K4me3 were observed in PMCs compared with somatic cells in the anther walls. This distinct chromatin modification pattern is similar to that in the MMC [52].

A whole-genome bisulfite sequencing (WGBS) analysis of male meiocytes showed high levels of CG and CHG methylation of TEs, similar to those in microspores and SCs [56]. In the CHH context, meiocytes and SCs have low methylation levels compared with somatic cells and especially VC. Male meiocytes have even lower CHH methylation level than microspores and SCs. Thus, CHH methylation increases upon male sex lineage development. Regions that were strongly hypermethylated in sex cells tended to be hypermethylated in all types of male sex lineage cells. This hypermethylation was most prominent in the CHH sites but involved CG and CHG sites as well. Of about the 1300 identified differentially methylated regions (DMRs), 97% were hypermethylated in sex cells compared with somatic cells. These sex lineage-specifically hypermethylated DMRs, were typically small (529 nt on average), collectively encompassing about 0.6% of the nuclear genome. Most of these DMRs (99.4%) were hypomethylated in male sex lineage cells in mutants affecting RNA directed DNA methylation (RdDM) pathway genes, namely *drm1 drm2* and *rdr2*. Hence, RdDM pathway activity is responsible for the establishment of these hypermethylated DMRs in male sex lineage genes. A significant share (42%) of these RdDM-dependent DMRs were sex lineage-specific, i.e., did not show non-CG methylation in somatic cells. Canonical sex lineage hypermethylated DMRs, which were non-CG-methylated in the somatic cells, appeared to be mostly located near TEs, but overlapped genes more frequently than other RdDM targets. Unexpectedly, most sex lineage-specific DMRs overlapped genes and were less likely to overlap TEs than randomly selected sequences of the same length. Hence, canonical sex lineage-hypermethylated DMRs appeared to be a subset of conventional TE-proximal RdDM targets, whereas sex-lineage-specific DMRs are a result of RdDM expansion into gene sequences. These last DMRs might repress the expression of the respective genes in sex lineage cells. Indeed, among the 47 genes that changed the expression more than four-fold in *drm1 drm2* compared with wild-type meiocytes, all were more active in mutant cells; seven of them overlapped a sex lineage-specific DMR, and one had such DMR within 20 bp. This is a much higher fraction (17%) than expected by chance (0.9%). Four of these genes were differentially expressed between meiocytes and leaves, and all of them were suppressed in meiocytes compared with leaves. The expression levels of these genes in leaves were not increased in RdDM mutants, indicating that RdDM specifically regulates their expression in meiocytes. Interestingly, certain pre-tRNA genes appeared to be among the preferred targets of sex lineage-specific RdDM. Totally, the 84 hypermethylated sex lineage-specific DMRs overlapped 100%, 75%, 73% and 42% of the phenylalanine, valine, cysteine, and methionine pre-tRNA genes, respectively. Consistently, 24 nt siRNAs were enriched at these pre-tRNA genes in pollen but not in shoots. The preferential hypermethylation of these pre-tRNA genes suggests that tRNA turnover may have some particular aspects in sex lineage cells. Since one DMR-covered methionine pre-tRNA gene is located within the last intron of *MULTIPOLAR SPINDLE 1* (*MPS1*) gene, the possible effect of its methylation on the *MPS1* mRNA splicing was investigated. Indeed, 28% of incorrect splicing events of *MPS1* at the last intron were detected in *drm1 drm2* mutant meiocytes, whereas none such splicing errors were observed in the wild type. Thus, methylation of the DMR within the *MPS1* intron is required for correct splicing of the *MPS1* transcript in meiocytes. This finding looked the more intriguing, as the *MPS1* gene is known to be necessary for meiosis, and retention of the last intron introduced a premature stop codon that disrupts the functional activity of the MPS1 protein. Indeed, the error-prone splicing of the *MPS1* mRNA in *drm1 drm2* and *rdr2* mutants was shown to result in a significantly higher occurrence of cellular triads and pentads that were never observed in wild type meiocytes. Hence, loss of methylation at the DMR within the last intron of *MPS1* causes the production of aberrant MPS1 molecules that interfere with meiosis.

## 4. Epigenetic Regulation at the Post-Meiotic Stage of Gametophyte Development

### 4.1. The Male Gametophyte

In *Arabidopsis*, mature pollen grains at the point of release from the anthers consist just of three cells. Comparison of the gene expression profiles between mature pollen grains and purified SCs indicated that VC and SCs have distinct transcriptomes. Compared with different sporophyte tissues, mature pollen has a smaller transcriptome (6587 genes expressed) with greater proportions of selectively (11%) or preferentially (26%) expressed genes [57]. In gene ontology categories, pollen transcriptome is skewed toward signaling, vesicle transport, and the cytoskeleton, suggestive of a commitment to pollen tube germination and growth. Comparative analysis of the transcriptomes between fluorescence activated cell sorting (FACS)-isolated SCs, pollen, and sporophyte tissues showed that SCs have a distinct transcriptional profile functionally enriched for DNA repair, ubiquitin-mediated proteolysis, and cell cycle regulation [58]. Of 53 genes known to be involved in small RNA pathways, RdDM, maintenance DNA methylation, and active demethylation, 18 genes were found to be expressed in SCs. Transcripts of five of these genes (*AGO9*, *DDM1*, *DRB4*, *MET1*, and *SUVH5*) are so highly enriched in SCs that their presence in pollen seems to be exclusively from SCs. Besides the cytosine DNA methyltransferase *MET1*, several other transcripts implicated in maintenance DNA methylation are expressed in SCs. Notably, the chromatin-remodeling factor DDM1, an important assistant of MET1 in the maintenance of CG methylation, is highly enriched in SCs. Interestingly, *CMT3* expression was not detected in SCs, while that of the DNA methyltransferase *DRM2* was [58]. As was already noted above, DRM2 catalyzes de novo methylation of DNA at all sequence contexts via the RdDM pathway in conjunction with DRD1, plant-specific RNA polymerases IV and V, and AGO4/AGO6/AGO9 [23]. While the largest subunit of Pol V (*NRPE1*) is expressed in SCs, expression of its second largest subunit, *NRPD2*, was not detected. *AGO6* is expressed, but *AGO9* shows much higher expression, suggesting a special role for AGO9 in de novo DNA methylation in SCs. Regarding the central role of 24 nt siRNAs in de novo DNA methylation via the RdDM pathway, active expression of two factors of their genesis, RDR2 and HEN1, in SCs is quite expectable. On the other hand, the dicer nuclease DCL3 that plays a key role in 24 nt siRNA generation from long dsRNA molecules produced by RDR2 appears not to be expressed in SCs. Perhaps, DCL1 that is expressed in SCs plays a similar role in these cells. It was demonstrated that the alternative RDR6-RdDM pathway mostly operates in the tissues where DNA methylation patterns are newly established to be carried to the next generation, such as in floral meristem cells, to establish TE methylation before the development of reproductive organs and gametes [27]. Thus, SCs actively regulate the epigenetic state of their genome through specific pathways of de novo and maintenance DNA methylation. The increased expression of the dsRNA-binding protein DRB4, which functions in the tasiRNA pathway, and of the *MOM1* gene, which is involved in a DNA methylation-independent epigenetic silencing pathway, add further pieces to this fascinating puzzle [58].

Small RNA pathways are active in male gametophytes that express diverse small RNAs, including a large number of siRNAs that match known transposons [59,60,61]. Supposedly, these siRNAs protect the germ cell genome against transposon activation. However, in the VC that does not participate in genome transmission to the next generation, transposons are actively expressed and transpose [61]. New polymorphisms resulting from TE insertions were found in mature pollen. No new insertions were detected in progeny, showing that transpositions do not occur in the somatic lineage leading to PMCs, because otherwise the insertions would be present in the SCs and passed to the next generation. From the same reason, TE activation could not have occurred before haploid mitotic divisions that produce the VC and SCs. Obviously, the TE activation occurs in the VC, which does not contribute DNA to the next generation. Therefore, both TE expression and mobility occur in the VC. It was found that the chromatin remodeling ATPase DDM1 accumulates in SCs, but not in the VC. Analysis of DNA methylation of representative TEs in DNA samples obtained from mature pollen and leaf showed that these TEs were highly methylated in leaf and retained some DNA methylation in mature pollen in all sequence contexts (CG, CHG, and CHH) but lost some methylated sites in the promoter regions. In DNA samples isolated from purified SCs, DNA methylation levels were high in all sequence contexts, and in particular, non-CG methylation was increased compared with whole pollen and leaf. Hence, the reduced DNA methylation in the pollen must be in VC.

Interestingly, the reactivation of TEs in mature pollen is associated with the loss of most 24 nt siRNAs and the dramatic gain of some 21 nt TE siRNAs [61]. It is in good accord with downregulation of the 24-nt siRNA RdDM pathway, and normal expression of the miRNA and tasiRNA pathways in mature pollen noted above [58]. Reduced DNA methylation, transcriptional reactivation, transposition, and selective loss and gain of siRNAs of TEs in pollen are highly reminiscent of similar TE features in *ddm1* mutants. On the other hand, *DDM1* transcripts are readily detectable in mature pollen and enriched in SCs [58]. By using DDM1-GFP fusion constructions, DDM1 was shown to accumulate in SCs but be undetectable in VC [61]. Therefore, the TEs reactivation in the VC could be attributed, at least partially, to loss of *DDM1* expression. Both DDM1 and 24 nt TE siRNAs are involved in repressing the TE activity. Thus, their loss from mature pollen appears to be in accord with the accumulation of TE transcripts and TE transposition. The loss of DDM1 results in reactivation of the Athila family TEs in VC and accumulation of their 21 nt siRNAs in SCs [61]. It was suggested that by this mechanism, activation of TEs in the VC reveals their presence in the genome, and silences these same TEs in SCs to prevent their transposition and genome instability [61]. Using VC- and SC-specific sensor transgenes Martinez and coauthors [62] have shown that TE-derived 21 nt siRNAs are indeed produced and functional in VC. Silencing effects of these siRNAs were lost in mutants for AGO1 and AGO2, known to bind 21 and 22 nt siRNAs and target complementary TE transcripts. AGO1 protein was found to be present in the VC cytoplasm but not in the SCs, and *AGO2* mRNA was enriched in VC. On the other hand, AGO4, which is known to be involved in RdDM, was not required for VC-specific silencing of sensor transgene. In addition to AGO1 and AGO2, DCL4 was required for silencing. DCL4 is known to generate 21 nt siRNAs from double-stranded RNAs as part of the post-transcription silencing pathway. Importantly, in the somatic plant tissues, these DCL4-derived 21 nt siRNAs have been shown to play a role in cell-to-cell silencing [63]. In accordance with its activity in VC, DCL4 was found to be expressed in mature pollen exclusively in the VC. These data demonstrated that AGO1, AGO2, and DCL4 are present in the VC, where they degrade TE mRNAs to produce 21 nt siRNAs that target transcriptionally active TEs. Moreover, it was shown that 21 nt siRNA produced from a sensor transgene in VC targets another sensor transgene specifically expressed in SCs [62]. Thus, TEs activation in VC could indeed be a mechanism limiting TEs expression in SCs and future embryo.

Among known *Arabidopsis* miRNAs identified in mature pollen sRNAome, those that target various components of the epigenetic systems are of particular interest [60]; miR168 and miR403 target *AGO1* and *AGO2*, respectively, miR162 targets *DCL1*, miR773 and miR778 target methyltransferases DNA methyltransferase *MET2* and two histone methyltransferases *SUVH*, respectively. The major proportion of the identified miRNAs target transcription factors. Other miRNAs target genes that are involved in hormone response and general metabolism, or those with unknown function. These small RNAs were likely involved in epigenetic regulation of gene expression earlier in development of male gametophyte and may also be maintained in dehydrated pollen to direct future changes in transcript abundance during pollen tube growth and even post-fertilization. With a higher sequencing depth by Illumina technology, over 100 known miRNAs were identified in the male gametophyte throughout development [64]. Besides, multiple new miRNAs that showed preferential expression in microspores and could be involved in the regulation of early stages of the male gametophyte development were found. A total of 75 known miRNA families in pollen and 83 in purified sperm cells were identified in *Arabidopsis* by Illumina sequencing of small RNA [61,65]. Their expression was mostly different between total pollen and purified SCs. For example, a sperm-specific miRNA family, miR159, is thought to regulate several transcripts of the MYB family transcription factors, such as *DUO1*, responsible for the expression of several germline genes [66]. Generally, SCs and VC appear to activate distinct miRNA pathways that correlate with their different cell fates and gene expression profiles.

The first indications of different chromatin regulation between VC and SC nuclei in male gametophyte came from microscopic and fluorescence in situ hybridization studies. A large-scale constitutive heterochromatin decondensation was observed in VC but not in SCs [67]. It correlated with a gross loss of the centromere chromatin-specific epigenetic marks CENH3 and H3K9me2 in VC but not in SCs. Unexpectedly, nearly complete CG methylation and intermediate CHG methylation were observed in the centromere-proximal *180CEN* repeat and Athila transposon sequences in both SCs and VC. Moreover, a marked increase in CHH methylation and a significant increase in CHG methylation were observed in VC compared with SCs. Thus, the heterochromatin decondensation in VC occurs with a concomitant loss of H3K9me2, retaining the heavy CG and CHG methylation, and significant gain of CHH methylation.

In comparison with the nine variants of histone H3 detected in somatic cells, only a limited subset of H3 variants is expressed in male gametophyte in *Arabidopsis* [68]. The VC expresses the canonical H3.3 variants HTR5 and HTR8, and the unusual variant HTR14, while the SCs express HTR5, the male germline-specific unusual variant HTR10, and divergent centromeric variant CENH3. Transcripts of H3.1 genes were not detected. Thus, the male gamete chromatin becomes distinct from the chromatin of the VC during pollen development.

A WGBS study of DNA methylation in purified microspores, VC and SCs showed a strong enrichment of DNA methylation at CG and CHG sites in the pericentromeric heterochromatin in pollen similar to methylation profiles in somatic cells, while CHH methylation in microspores and SCs was lost from pericentromeric retrotransposons and satellite repeats but subsequently restored in the VC [69]. In pairwise comparisons between cell samples, almost all DMRs were found in intergenic regions and transposable elements. Notably, almost all CHH DMRs were hypomethylated in SCs, whereas CG DMRs were hypomethylated in the VC. More than two thousand CHH DMRs overlapped with 1781 different TEs, including 1483 LTR/Gypsy elements and 139 DNA transposons. These TEs were similarly CHH-hypomethylated in microspores and SCs compared with VC.

On the other hand, 221 CG-hypomethylated DMRs were found in the VC compared with SCs that overlapped with 109 different TEs, including *AtMu1a*, in accordance with data cited above [61,70]. In contrast, CG methylation was very similar between SCs and microspores with only a few loci demethylated in microspores [69]. These same loci (15/21 DMRs) were also CHG-demethylated in microspores compared with VC and SCs. CG DMRs in the VC and CHH DMRs in SCs did not overlap, suggesting that their differential methylation might be due to the different activity of DNA methyltransferase and demethylase enzymes in each cell type. By pairwise comparisons of CG DMRs between VC and SCs in wild-type, *dme* mutant and *ros1 dml2 dml3* (*rdd*) mutant plants, of the 221 DMRs hypomethylated in the VC, 134 were shown to be targets of RDD, and 48—targets of DME. Since all DNA demethylase genes are known to be expressed in the VC and none of them are expressed in SCs [70], DNA demethylases should be responsible for the loss of CG methylation in the VC. The overall level of CHH methylation in microspores and SCs was much lower compared with the inflorescence as if reductional meiotic and/or haploid mitotic divisions were not accompanied by RdDM. Indeed, the DNA methyltransferase DRM2 necessary for RdDM was prominent in VC but barely detectable in microspore and SCs [69]. As has been noted above, other genes involved in 24 nt siRNA-dependent RdDM were also not expressed in SCs [58]. Since CHH methylation of transposons inherited through SCs is known to be restored in the embryo [19,71], this methylation must occur at fertilization. An accumulation of maternal 24 nt siRNAs was observed to occur in the seed coat and the endosperm [72,73]. Obviously, maternal 24 nt siRNA might target CHH methylation machinery to TEs inherited through SCs. Since many DMRs that lost CG methylation in the VC flank imprinted genes, methylation patterns in repeats flanking the imprinted maternally expressed gene *SUPPRESSOR OF DRM2/CMT3* (*SDC*) and paternally expressed type I MADS box gene *PHERES1* (*PHE1*) were investigated. It has been shown that *SDC* is expressed when flanking repeats are unmethylated [74], while *PHE1* is expressed when a tandem repeats downstream of the coding sequence are methylated [75]. Tandem repeats flanking both genes were found to lose CG methylation in the VC [69]. However, CHH methylation was detected at *SDC* but not at *PHE1*. It was correlated with accumulation in SCs of 24 nt siRNA complementary to *SDC* but not 24 nt siRNA complementary to *PHE1*. An even higher level of this siRNA was detected in total pollen, indicating that it could be produced in VC. Of 12 imprinted maternally expressed and 16 paternally expressed genes that had a TE within 2 kb of the coding sequence, all 28 TEs lost CG methylation in the VC compared with the progenitor microspore, but only those surrounding maternally expressed genes were targeted by siRNA and CHH methylation in pollen.

Interestingly, it was found that although most CG DMRs accumulate both 21 and 24 nt siRNA in SCs, those associated with imprinted genes accumulate only 24 nt siRNA. Furthermore, siRNA levels for maternally expressed genes were much higher than for paternally expressed genes in SCs and seeds but not in total pollen. Thus, 24 nt siRNAs from repeats surrounding maternally expressed genes accumulate preferentially in SCs. Similar to 21 nt siRNA [61], these 24 nt siRNAs may be derived from the VC. Collectively, the data described show that the maternally expressed genes are epigenetically silenced in SCs by DNA methylation and this silenced status is transferred into the next generation.

The DNA methyltransferase dependence of DNA methylation in the male gametophyte cells was studied in more details via the WGBS analysis of isolated SCs and VC nuclei from *cmt3*, *cmt2*, *drm1 drm2* mutant, and control wild type plants [76]. Globally, it was found that none of the mutations have a major effect on CG DNA methylation of genes and TEs, just like in somatic tissues. The *cmt3* mutation had a strong effect on CHG methylation of TEs in SCs and VC. Thus, this type of methylation is maintained mostly via CMT3-H3K9me2 pathway both in SCs and VC, despite the gross loss of H3K9me2 in the VC noted above [67]. Mutation of either *cmt2* or *drm1 drm2* affected the patterns of CHH methylation similarly in SCs and VC, though CHH methylation is much higher in VC compared with SCs [69,77]. Similar to somatic tissues, in SCs and VC, CHH methylation in H3K9me2-poor chromatin TEs mostly depended on DRM activity, CHH methylation in TEs with intermediate H3K9me2 levels—both on DRM and CMT2, and methylation of the most H3K9me3-rich chromatin TEs—on CMT2. Of note, VC appeared to be modestly higher CHH-methylated in euchromatic (H3K9me2 poor) TEs, but much higher—in heterochromatic (H3K9me2 rich) TEs, compared with SCs. Hence, the reported elevated CHH methylation in the VC is mostly caused by increased H3K9me2-dependent CMT2 methylation in heterochromatin.

Interestingly, the effects of *cmt2* mutation on CHH-methylation appeared to be significantly weaker in the VC compared with both SCs and somatic tissues. Heterochromatic loci that retained CHH methylation in *cmt2* mutant VC are targeted both by CMT2 and DRM, very much like TEs in chromatin with intermediate levels of H3K9me2. It should be noted, however, that such loci represent less than 8% of those targeted by CMT2. As expected, CHH methylation of these loci in somatic tissues depends on CMT2 activity, but it becomes RdDM-dependent in *ddm1* mutant plants, as evidenced by its reduction in combined mutants *ddm1 rdr2* or *ddm1 drd1*. Heterochromatic TEs targeted by RdDM in the VC appeared to be more likely to be transcribed in *ddm1* mutant somatic tissues than other heterochromatic TEs. Overall, these data support the view of RdDM targeted heterochromatic TEs to be transcriptionally activated in the VC [61]. Methylation of individual CG sites in heterochromatic TEs and genes and euchromatic TEs were found to be much higher in microspore, SCs, and VC than in somatic tissue [76]. Since only germ cells contribute to the next generation, their higher methylation efficiency could be a special mechanism to ensure the stable maintenance of DNA methylation patterns in the offspring.

In 4′,6-diamidino-2-phenylindole (DAPI)-stained semi-thin sections of rye bicellular pollen, the GC nucleus looks more condensed and smaller than VC nucleus [78]. In nuclei of GC and SCs, the enhanced density of histone H3K4me2 is observed towards the nuclear membrane, while the VC nucleus shows a weak intensity and dispersed distribution of this mark. H3K9me2 mark shows an almost uniform distribution throughout generative nuclei, typical for heterochromatin in plants with large genomes. VC nucleus of bi- and tricellular pollen has less H3K9me2 than GC and SCs. Clusters of H3K27me3 are observed at the telomeric poles in VC nucleus, while generative nuclei contain a small quantity of H3K27me3 only in bicellular pollen. Only GC and SCs nuclei contain detectable H3K9ac, while VC nucleus is H3K9ac-negative. The active (phosphorylated) large subunit of RNA polymerase II is homogeneously distributed throughout the nucleoplasm in VC nucleus, while GC and SCs nuclei are almost devoid of it. These data show that the transcription activity of GC and SCs of rye pollen is strongly reduced compared with the VC nucleus.

### 4.2. The Female Gametophyte 

The female gametophyte cell specification is also epigenetically regulated. Haploid mitotic divisions of the megaspore depend on mobile siRNAs produced in surrounding cells since its developmental arrest occurs in *ago5* mutant [43]. After cellularization, chromatin configurations are different between the EC and CC. The maturing CC expresses a broader set of histone H3 variants than the EC [68]. The CC expresses H3.1 and H3.3 from several *HTR* genes each and the unusual variant HTR14. Upon maturation of the CC, the histone H3.3 encoded in *HTR5* gene gradually declines, while that encoded in *HTR8* gene increases. In contrast, the EC does not contain detectable H3.1 variants, although low levels of *HTR1* and *HTR3* transcripts are detectable. During EC maturation, H3.3 variant encoded in *HTR5* gene increases, while that encoded in *HTR8* gene decreased to undetectable levels in mature EC. The unusual H3 variant HTR14 was detected in the CC only. Unexpectedly, although EC chromatin maintains chromocenters, neither CENH3 protein nor *CENH3* transcripts were detected in EC. In contrast, the CC chromatin does not contain visible chromocenters, which is in accord with CENH3 protein absence, but *CENH3* transcripts are abundant in CC. Thus, the EC chromatin contains only H3.3 at maturity. Very reminiscent of pollen VC, a severe loss of heterochromatin and H3K9me2 occurs in the CC [53]. This change correlates with the transcriptional activation of TEs [79]. Unlike the CC, the EC contains a high amount of heterochromatin marked by H3K9me2 [53]. Mutations in *DML* genes prevent the heterochromatin loss in the CC but do not affect the heterochromatin in EC [53]. Obviously, DNA demethylation causes the heterochromatin loss in the CC. Conversely, the *cmt3* mutation has no impact on the CC chromatin organization but leads to the complete loss of H3K9me2 in the EC. Notably, this effect is detectable in post-cellularization ovules only, indicating that CMT3 activity is necessary to establish or maintain H3K9me2 profile in the EC following cellularization. Thus, different mechanisms regulate the distribution of the repressive chromatin mark H3K9me2 in the two female gametes. These mechanisms act via opposite changes in DNA methylation, a hypomethylation in transcriptionally active CC and hypermethylation in quiescent EC. To a certain extent, these distinct chromatin states in the CC and EC are transferred post-fertilization into endosperm and embryo, respectively [53].

No expression of DNA methyltransferases is observed in the CC, probably contributing to the heterochromatin loss. The expression of the major maintenance DNA methyltransferase MET1 was shown to be barely detectable in EC and absent in CC of mature female gametophyte [19]. On the other hand, DNA methyltransferases MET2a and MET2b are expressed in CC but not in EC, whereas MET3 is not expressed in any part of the mature ovule. Thus, no CG-specific DNA methyltransferase appears to be present in EC. The transcriptome sequencing analysis of isolated EC and SyCs showed that indeed there is some expression of *MET1* at a rather low level in EC but not in SyCs. Relatively high levels of the CHG-specific DNA methyltransferase *CMT3* mRNA were detected in EC but not in SyCs. The expression of its homologs *CMT1* and *CMT2* was not detected in any part of the mature female gametophyte. In mature ovules, the major DNA methyltransferase of de novo type DRM2 is expressed in the ovule integuments and both female gametes, in EC at a higher level than in CC. DRM1 expression was not detected in somatic tissues and ovule integuments, whereas its active expression was detected in the mature EC. Thus, among all DNA methyltransferases, the EC expresses predominantly de novo type enzymes DRM1 and DRM2. In contrast, the CC and the SyCs express low levels of de novo DNA methyltransferases and do not express maintenance DNA methyltransferases. These features could explain the expression in CC of genes that are silenced by DNA methylation in somatic tissues. No expression of MET1, DRM2, and CMT3 was detected in the endosperm in contrast with the high levels observed in the seed coat. Unexpectedly, DNA methyltransferases MET2a, MET2b, MET3, and CMT2 were found to be actively expressed in the endosperm [19,80]. Together with the expression of the DNA demethylase DME in the CC [29], the reduced expression of major DNA methyltransferases in the CC and endosperm could be the cause of reduced DNA methylation in the endosperm [19,71,81].

All three major DNA methyltransferases, MET1, DRM2, and CMT3, are expressed at high levels in the embryo [19]. A bisulfite sequencing analysis of five loci known to be silenced by DNA methylation in adult plants (*AtSN1*, *FIS2*, *FWA*, *MEA*, and *SDC*) showed no changes in CG and CHG methylation throughout embryo development, while CHH methylation gradually increased. This CHH methylation is lost in the double mutant *drm1 drm2* plants and mutants for RNA polymerases IV and V, indicating that the RdDM pathway is responsible for gradual de novo CHH methylation observed in the developing embryo. During early embryogenesis, the decrease in CHH methylation in the *drm1 drm2* double mutant is larger than in single mutants *drm1* and *drm2*. Later, no impact of *drm1* on CHH methylation is detectable. No expression of DRM1 was detected at all stages of embryo development. Apparently, DRM1 activity that contributes to CHH methylation in early embryos is derived from its expression in the EC. In the late embryo and adult plant tissues, DRM1 is no longer expressed, and DRM2 becomes the only de novo methyltransferase in the RdDM pathway. A paternally transferred unmethylated *FWA-GFP* reporter transgene from a *met1*/+ mutant pollen was continuously expressed in endosperm but gradually lost expression in the embryo by 72 hr after pollination [19]. Since silencing of *FWA-GFP* requires methylation in CG and CHH contexts, and MET1 and CMT3 are unable to methylate completely demethylated paternal alleles, the complete silencing of *FWA-GFP* implies that RdDM would methylate CG, CHG, and CHH de novo in the embryo. The data described show that in the EC, DNA methylation is carried out mainly by de novo DNA methyltransferases DRM1 and DRM2. These DNA methyltransferases together initiate de novo DNA methylation in the early embryo to silence loci sensitive to DNA methylation. This de novo DNA methylation is dependent on 24 nt siRNAs that might be derived from male gametes, maternal ovule tissues, or the endosperm. Otherwise, the embryo itself may produce these 24 nt siRNAs.

Using fluorescent sensors capable of specifically detecting CG or CHH methylation at single-cell resolution, Ingouff and coauthors [82] showed that CG methylation is stable across all stages of female sporogenesis and gametogenesis but decreases to a lower level in mature EC. Following fertilization, a typical level of CG methylation was restored in the embryo. In contrast to the relative stability of CG methylation, CHH methylation became undetectable in the MMC at the early stage of sporogenesis and gradually reappeared at later stages. These changes correlate with the establishment of a distinct chromatin structure devoid of the CENH3 and linker H1 histones [52]. On the other hand, CHH methylation was stable throughout female gametogenesis, including the mature EC. Of non-CG DNA methyltransferases CMT2, CMT3, and DRM2, only DRM2 appeared to be absolutely necessary for CHH methylation in the mature EC, which is in agreement with the fact that DRM1 and DRM2 are the only DNA methyltransferases present in the mature EC [19]. Unexpectedly, of two RNA polymerases involved in 24 nt siRNA-dependent RdDM pathway, only Pol V appeared to be necessary for CHH methylation in the EC, whereas Pol IV was dispensable. In accord with this notion, NRPE1 (Pol V-specific catalytic subunit) and NRPD2 (Pol IV-V common catalytic subunit) were detected in the EC, whereas CHH-specific DNA methyltransferase CMT2 and Pol IV-specific catalytic subunit NRPD1 were absent. Thus, CHH methylation in the EC occurs via an alternative RdDM pathway that is dependent on the activity of DRM2 and Pol V but does not depend on the activity of Pol IV.

The genome-wide DNA methylation patterns were analyzed in pure samples of the *Arabidopsis* CC nuclei by WGBS [83]. The overall patterns of DNA methylation in genes and TEs were virtually identical between the CC and other cell types and tissues. Genes and TEs were found to be extensively methylated in CC at CG, with methylation levels slightly lower than in the male gametophyte SCs or embryos but higher than in endosperm. CHG methylation of TEs appeared to be similar to embryos and male gametes and substantially higher than in endosperm. CHH methylation patterns of TEs in CC resemble those in SCs, with levels higher than those of SCs but lower than VC or embryos. The non-CG methylation patterns of neither SCs nor CC closely resemble those of endosperm, indicating that DNA methylation is remodeled after gamete fusion. CG methylation of most loci was similar between SCs and CC, but there were some loci hypomethylated in the CC. Almost all loci that show demethylation of maternal chromosomes in the endosperm are hypomethylated in the CC. Similar results were obtained with loci hypomethylated in the VC, but with fewer sequences compared with CC, in accord with reported partial overlap between endosperm and VC demethylation [77]. Although CC showed hypomethylation of the vast majority of loci demethylated in the endosperm, the extent of this hypomethylation was lower compared with the maternal endosperm chromosomes and the VC [83]. Patterns of CG methylation were similar between CCs from wild-type and *dme*/*+* heterozygous mutant plants, except for loci known to become demethylated on maternal alleles in endosperm; methylation levels of these loci were higher in mutant plants. Thus, the DME-dependent demethylation on maternal endosperm chromosomes appears to be initiated in the CC.

In rice plants, the methylation patterns of CC and EC were found to be similar to those of other tissues in genes and TEs [83]. CG methylation levels in the CC are lower than those in embryos, roots, and leaves, resembling the methylation in the endosperm, whereas CG methylation in the EC is slightly higher compared with other cells. CHG methylation in the CC TEs is slightly lower compared with embryos, roots, and leaves but much higher than in endosperm, while CHG methylation in the EC is much higher than in other tissues. CHH methylation in rice is concentrated in small, gene-adjacent TEs, a feature shared with maize [84]. Compared with somatic tissues, CHH methylation is very low in endosperm and much higher in the CC and EC. In EC, high CHH methylation is also present in long TEs. The levels of non-CG methylation in the CC are not predictive of those in the endosperm, and those in the EC are not predictive of those in the embryo. CG methylation is similar between the CC and EC, but there is a substantial group of loci relatively hypomethylated in the CC. Similar to *Arabidopsis*, most loci that show demethylation of maternal chromosomes in the endosperm are hypomethylated in the CC, while the extent of their hypomethylation in the CC is lower than in endosperm. Hence, the DNA demethylation of maternal endosperm chromosomes observed in different plants is initiated in the CC.

Similar to the transport of siRNAs occurring between VC and SCs in the male gametophyte, it has been suggested that siRNAs can move from the CC to the EC to silence TEs [77]. The CC and EC appear to remain symplastically connected even after cellularization. It was shown that 1 h after microinjection in CC, fluorescently labeled 24 nt sRNA is detectable in EC [85]. About 8850 genes were found to be expressed in laser microdissected female gametophyte, a significant share of them being specifically enriched in one cell type only [86]. Relative to other cell types, high expression of genes encoding the double-stranded RNA-binding factors (DCL1, HYL1, and AT4G00420) and RISC components such as AGO1 occurs in the EC. On the other hand, SGS3, involved in posttranscriptional gene silencing, is predominantly expressed in the CC. These findings are in accord with the view that RNA-based silencing plays an important role in the female gametes.

Globally, male and female gametophytes show similar transcriptome features that are distinct from those of sporophytic tissues. Of note, the embryo transcriptome appeared to be more similar to those of gametophytes than to the adult sporophyte tissue transcriptomes. About 400 genes show significant enrichment in one of the female gametophyte cell types. Some of these differentially expressed genes are involved in gametophyte development, such as *MYB98* in SyCs and *FIS2*, *DME*, *CKI1*, *UNE6*, and *EDA28* in the CC. Gene sets involved in transcriptional, posttranscriptional, and epigenetic regulation, signaling, and cell wall modification are enriched in the whole gametophyte. Three groups of transcription factors are also overrepresented, namely the RWP-RK (RWPYRK) domain, the MADS (MCM1, AGAMOUS, DEFICIENCE, SRF) domain type I, and the reproductive meristem families. MADS-domain type I proteins are exclusively enriched in male and female gametophytes, and developing embryo. Of the 28 MADS-domain type I genes studied, seven are maximally expressed in the female gametophyte, including *AGL62* that is known to play a role in endosperm cellularization and shows the highest expression in the CC. Other family members are maximally expressed in the male gametophyte (10 genes), the embryo (9 genes), or seed (2 genes). These data suggest a predominant role of MADS-domain type I transcription factors in sexual reproduction. Genes encoding PAZ, Piwi, and DUF1785 domains, mainly associated with ARGONAUTE and DICER proteins, are globally enriched in the EC. Enrichment for small RNA pathways is a prominent feature of the generative female gamete (EC), similar to the male gametes (SCs). This feature could be important for gamete and future embryo protection against transposable elements. Endosperm DNA of *Arabidopsis thaliana* is known to be globally demethylated compared with embryo DNA in all contexts [71]. The DME DNA demethylase is highly expressed in the CC before fertilization and is required for the endosperm DNA demethylation at the maternal chromosomes inherited from the CC. Passive demethylation caused by downregulation of the maintenance DNA methyltransferase MET1 in CC also contributes to the maternal endosperm genome undermethylation [87]. Both slight global hypomethylation and strong local demethylation at the CG sites were observed in the endosperm maternal genome compared with the paternal genome [77]. The local demethylation was absent in *dme* mutant endosperm. Thus, DME is the major DNA demethylase active in the CC. This active demethylation affected at least 9816 specific sequences to account for the methylation differences between the maternal and paternal genomes in the endosperm. Obviously, the differential expression of the epigenetic pathway components between embryo and endosperm could be already established in the EC and CC.

### 4.3. Common Epigenetic Mechanisms in Male and Female Gametophytes

An RNA-seq analysis of transcriptomes in VC, SCs, and EC of rice showed these transcriptomes to be highly divergent [88]. Approximately 75% of all genes assessed among the three cell types were found to be differentially expressed. In the pairwise comparisons, 14,236 genes were found to be equally expressed between EC and SCs, 20,650 genes between SCs and VC, and 16 627 genes between EC and VC. It was found that the histone variants expressed in the EC are distinct from those expressed in the SCs and VC. As was noted above, the chromatin of plant SCs is highly condensed and contains an SC-specific histone H3 variant HTR10 (MGH3) [68]. Rice has one putative HTR10 ortholog that was found to be preferentially expressed in the SCs. However, the histone H3 variant with the highest expression level in mature SCs was not HTR10 but a variant histone H3.3—a replication-independent histone typically associated with euchromatin [88]. The histone H2A variant, H2A.Z, is known to accumulate at promoters that are poised for transcription. Rice has three putative H2A.Z orthologs; two of them were found to be enriched in EC, consistent with the post-fertilization transcription of the maternal genome in the zygote. Notably, the third H2A.Z gene is actively expressed in SCs. Probably some paternal genes are poised for expression in the zygote.

Both EC and SCs were found to express the genes encoding all four components of PRC2 [88]. The EC shows expression of all possible subunits of the PRC2 except for FIE1, while SCs express a more limited subset consisting of MSI1, FIE2, CLF, and EMF2B. Thus, both female and male gametes are capable of PRC2-mediated gene silencing via histone H3K27me3-methylation. Genes encoding the JMJ class histone demethylases are widely expressed in EC and SCs, with slightly lower expression in the VC. Notably, the SCs were found to express high levels of ZOS1-20–the putative rice ortholog of the *Arabidopsis* H3K27 demethylase REF6. Of four genes encoding H3K4me3-specific LDL demethylases, all are expressed in the EC, whereas only *LDL3* expresses at moderate levels in SCs. Of ten DNA methyltransferase genes found in rice, low expression of *MET1* was detected in EC and SCs and nearly undetectable expression in VC, whereas *OsDRM2* is expressed in SCs and at a low level in VC. Both copies of *DDM1* are expressed at a lower level in the VC than in the SCs. Rice contains two orthologs of the DNA demethylase DML3, four orthologs of ROS1, and no orthologs of DME. One of the ROS1 orthologs, *ROS1a*, appears to be required for the proper development of gametes since a null-mutant allele of this gene is not transmitted to the progeny [89]. In accord with this view, the *ROS1a* gene was found to be the most highly expressed *ROS1* gene in all cell types, EC, SCs, and VC [88]. Of major RNA interference pathways genes, the miRNA-specific genes *DCL1* and *AGO1* were found to be expressed in all three cell types, whereas *HEN1* was expressed at a moderate level in the EC, at a very low level in the SCs and was nearly undetectable in the VC. Thus, the miRNA pathway could be potentially active in the EC only. Of the 21 nt siRNA-dependent PTGS pathway genes, *RDR6*, *DCL4*, *AGO1,* and *AGO7* were found to be expressed in the EC, whereas the SCs and VC lacked *RDR6* and showed barely detectable expression of *DCL4* and *AGO7*. Of the 24 nt siRNA-dependent TGS pathway genes, only *RDR2* and *AGO4* showed expression in the EC, whereas *DCL3* was expressed in all three cell types. Thus, the activity of the canonic 24 nt siRNA pathway in the SCs is unlikely, but an alternative entry pathway could potentially play a role in TE silencing in the female gametes.

The *Arabidopsis*, DEMETER (DME) DNA demethylase was shown to be primarily expressed in the CC of the female gametophyte before fertilization [29] and to be required for maternal allele DNA demethylation in the endosperm that establishes gene imprinting [90]. This selective DME expression globally reprograms the DNA methylation landscape of endosperm, including demethylation of repeated sequences and TEs [71,81]. A maternally inherited null-mutant *dme* allele results in inviable seeds in all *Arabidopsis* ecotypes, while the paternal *dme* alleles are efficiently transferred to progeny in the Ler ecotype but show decreased viability in Col [70]. *DME* expression was detected in Col and Ler pollen, but not SCs, suggesting that it is expressed in the VC. Moreover, DME activity was required for demethylation of the *MEA* and *FWA* genes and a *Mu1a* transposon in the VC in Col and Ler pollen. Hence, DME demethylates similar groups of genes and transposons in the VC of the male gametophyte and the CC of the female gametophyte. Since the VC genome does not participate in double fertilization, DME activity in the VC, unlike its activity in the CC, does not affect gene imprinting in the endosperm. Nevertheless, DME activity in the VC may be important for male fertility, probably by contributing to the methylation landscape reprogramming in the VC. The transcripts of other DNA demethylase genes (*ROS1*, *DML2*, and *DML3*) were detected in pollen but not in pure SCs. Nevertheless, no changes in DNA methylation in VC and SCs were detected in homozygous triple mutant *ros1 dml2 dml3* stamens. Thus, DME seems to be the only DNA demethylase that affects VC genome demethylation.

A comparative study of DNA methylation patterns between FACS-purified VC and SC nuclei showed that at least 9932 loci are specifically demethylated in the VC [77]. Interestingly, these VC-specifically demethylated CG sites overlap by about 45.5% with those demethylated in the maternal endosperm genome and, probably, in the CC. In heterozygous *dme*/*+* plants, half of the pollen lacks DME [70]. CG sites that are demethylated in VC show higher methylation level in VC isolated from *dme*/*+* pollen, indicating that DME activity is responsible for their demethylation in VC [48]. CHG methylation is generally higher in the VC than in the SCs, but loci demethylated at CG sites are also demethylated at CHG sites—another feature similar between the VC and CC. CHH methylation is very high in the VC, and low in SCs, consistent with the strong RdDM activity in the VC noted above [67]. Nonetheless, CG-demethylated loci tend to be less CHH-methylated in VC than in SCs. Obviously, a similar active DNA demethylation by DME activity occurs in both types of gametophyte companion cells. Local demethylation in VC and CC mostly affects smaller AT-rich TEs that are enriched for euchromatin-associated histone modifications, consistent with the decondensation of heterochromatin observed in VC [67] and CC [53]. Small TEs tend to occur near genes, explaining the observation that CHG and CHH methylation near genes is lower in wild-type VC compared with SCs, though overall non-CG methylation is higher in VC than in SCs [77]. In view of the hypothesis that DNA demethylation and activation of TEs in the VC generates siRNAs that would reinforce silencing of complementary TEs in SCs [61], such TEs could be demethylated in the CC, and their maternal copies should remain active in the endosperm. Indeed, 11 TEs demethylated by DME that are specifically maternally expressed in wild-type, but not *dme*, endosperm were identified [90]. It was also demonstrated that sRNAs could travel from the CC to the EC [77]. If the VC and CC siRNAs induce the silencing of complementary loci in SCs and EC via RdDM mechanism, lack of DME in the VC and CC should reduce the methylation of DME target sequences in SCs and EC. Indeed, overall CHH methylation of TEs was decreased in *dme*/*+* SCs compared with wild-type SCs, and the loci CG-demethylated by DME in VC were preferentially CHH-hypomethylated in *dme*/*+* SCs [77]. Conversely, loci that exhibit decreased CHH methylation in *dme*/+ SCs showed increased CG methylation in *dme*/*+* VC. Thus, DME activity in the VC is required for full methylation of TEs in SCs.

It was shown that ARID1, an AT-rich interacting domain transcription factor, is required for TE silencing and is specifically expressed in the GC of bicellular pollen, but only in VC in mature pollen [91]. Furthermore, it was shown that the *ARID1* gene expression is negatively regulated by MET1-dependent CG methylation [92]. Localization of ARID1 was identical between wild type and *met1* plants in microspore and bicellular pollen, whereas in mature pollen ARID1 was confined to VC in wild type but present both in VC and SCs in *met1* mutant plants. Hence, MET1 activity appears to repress the *ARID1* expression in SCs of wild type plants. In contrast, ARID1 did not affect MET1 distribution in mature pollen. In female gametophyte, ARID1 was expressed in the MMC but was relatively confined in the CC in the mature female gametophyte. Thus, ARID1 is preferentially confined to companion cells in both male and female gametophytes. Subcellular localization of MET1 in female gametophyte was similar between wild type and *arid1* mutant plants before maturity, but ectopic MET1 expression in the CC was observed in *arid1* mutants. Thus, ARID1 conversely downregulates MET1 expression in the CC in the female gametophyte.

## 5. Conclusions

Both stages of gametophyte development involve extensive epigenetic reprogramming, including siRNA dependent changes in DNA methylation and chromatin restructuring. These events could play a fundamental role in the derepression of gamete-specific genes, maintaining genome integrity, and setting the epigenetic landscape for seed development [93,94]. Furthermore, this intricate mosaic of epigenetic changes determines, to a great extent, embryo and endosperm development in the future sporophyte generation. The massive activation of TEs in companion cells that do not contribute DNA to the next sporophyte generation in both male (VC) and female (CC) gametophyte serves as a means to deliver the respective siRNAs into gametic cells (SCs and EC) in order to silence TEs and preserve the genome of the future embryo.

## Figures and Tables

**Figure 1 ijms-20-03051-f001:**
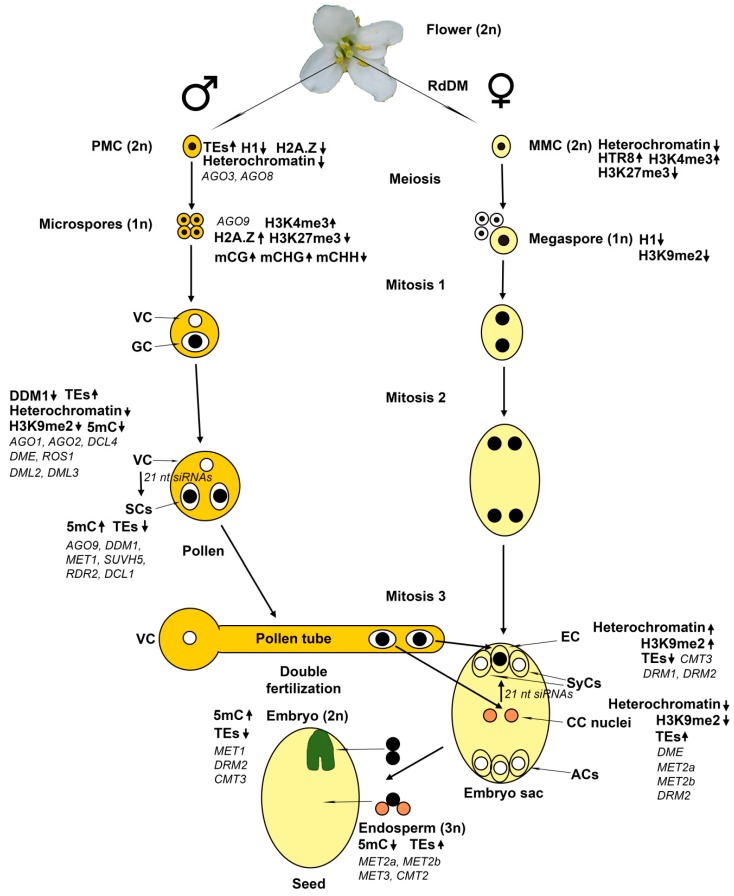
Key events in plant gametogenesis. PMC—pollen mother cell, MMC—megaspore mother cell, RdDM—RNA directed DNA methylation, VC—vegetative cell, GC—generative cell, SCs—sperm cells, EC—egg cell, SyCs—synergid cells, CC—central cell, ACs—antipodal cells. Increases or decreases in epigenetic features are shown by arrows pointing upwards or downwards, respectively. Genes actively expressed are shown in italics.

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
