# Peer review of "Epigenetic Regulation of Plant Gametophyte Development"

_ijms, 2019, doi:10.3390/ijms20123051_

Round 1
Reviewer 1 Report
The review of Ashapkin and coauthors deals with a critical stage of plant reproduction, the production of haploid gametes by a special haploid generation - the gametophyte. In flowering plants, this haploid generation is highly reduced to male and female gametophytes consisting of only a few cells each. Both gametophytes are produced within the somatic tissues of the diploid (sporophyte) generation. These features are probably the main reasons why the knowledge of the epigenetic mechanisms of gametophyte development still remains relatively poor compared with those of sporophyte development.
In their paper, the authors analyze the data concerning this subject accumulated up to the present time. The paper is rather comprehensive and demanding at some places, such as discussion of the WGBS data on male meiocyte DNA methylation, comparisons between small RNA pathways active in different gametophyte cells, the possible role of gametophyte epigenomes in embryo and endosperm developments and some others. Undoubtedly, these data could be of great interest for the wide audience of the IJMS, not only involved in epigenetic studies.
It would be very helpful if the authors add figures depicting stages of gametophyte development and key events of its epigenetic regulation. After this small additions, the paper could be published in IJMS.
Author Response
We have added a figure depicting stages of gametophyte development and key events of its epigenetic regulation.
Reviewer 2 Report
The authors organized and evaluated recent primary literature focusing on the epigenetics of plant gamete development. The review is timely and covers a fascinating and important field. The manuscript is subdivided into 1) an introductory paragraph describing the biology of gamete formation, starting with the respective mother cells up to fertilization, 2) a paragraph summarizing epigenetics during meiosis, and 3) a final paragraph describing epigenetics after meiosis. The review caters to plant epigenetics experts wishing to grasp recent developments in epigenetic regulation of gamete formation in plants.
The review is timely and covers a priority subject in plant development/epigenetics. The manuscript is a well-thought-out narrative review, integrating the relevant primary literature, suitable in quantity and timeliness. Space is taken to elaborate on the literature evidence, using precise numbers. The manuscript title and abstract are informative.
I do not have major concerns and can imagine seeing the review published. Nevertheless, I recommend a couple of changes, which may positively impact usefulness to the reader and, hence, citation rate:
(1) Introduction: The review gives a thorough introduction on plant gamete formation. However, regarding epigenetics, the manuscripts jumps in directly. I suggest integrating an additional chapter, shortly (!) touching on the epigenetics components that are needed later on, such as DNA methylation, histone modification, siRNAs, RdDM, and its components such as the Ago family. Parts of the manuscript could be easily moved into this new chapter, e. g. the Ago family description (page 3, line 141 cont’d). This would make the manuscript more accessible for non-epigenetics experts, and invite readers with a plant development background.
(2) I enjoyed reading the review as it uses the narrative style to explain findings in detail. Nevertheless, reading an uninterrupted text is challenging, especially if multiple evidence is presented and compared. In addition to the narrative, the manuscript would benefit from a systematic overview of current findings addressing the impact of epigenetic regulation on gametophyte development. My proposition is to include at least one show item, either a table or a figure, better both, to summarize individual epigenetic layers/analyses per cell stage. A table could for example include: development stage, cell type, male/female, type of analysis (transcriptome, BS-seq, ChIP-seq…), major result, citation.
A figure could schematically show each step in gametophyte development including all relevant cell types. Epigenetic changes, e. g. reduction and rebuilding of heterochromatin (page 4 line 173 cont’d) could be directly marked. This approach would increase usefulness and citation of the article, and may lead to inclusion of the figure in lectures, seminars etc.
(3) It would be useful to also address open research questions at the end of the manuscript to make the text appealing for decision makers.
Minor changes:
(4) Page 2, line 73, “In cereals, ACs resemble those involved in nutrient transfer…” – What does “those” refer to?
(5) Page 10, line 497: “DNA transferase”, should be “DNA methyltransferase”
(6) Page 15, line 723: Why is “development” written in italics?
Author Response
We have added an additional section on the basics of epigenetic regulation in plants.
We have added a figure that shows key epigenetic events at every stage of gametophyte development.
As concerning the open research questions, we think that the main problem is not the missing experimental data but rather the uniting those already available into a general picture of gametophyte epigenetic regulation. Though of course, there would be a further accumulation of fine details along these lines that eventually will lead to more advanced knowledge of the problem as a whole.
In cereals, antipodal cells have morphological features typical of cells that accumulate and secrete large amounts of nutrient materials. Therefore, a nutritive role is the most popular proposal concerning their function.
Corrected
By pure accident. Corrected